# Fruit fracture biomechanics and the release of *Lepidium didymum* pericarp-imposed mechanical dormancy by fungi

Katja Sperber [1], Tina Steinbrecher [2], Kai Graeber[2], Gwydion Scherer [1], Simon Clausing[1], Nils Wiegand[1], James E. Hourston [2], Rainer Kurre[3], Gerhard Leubner-Metzger [2] & Klaus Mummenhoff [1]

The biomechanical and ecophysiological properties of plant seed/fruit structures are fundamental to survival in distinct environments. Dispersal of fruits with hard pericarps (fruit coats) encasing seeds has evolved many times independently within taxa that have seed dispersal as their default strategy. The mechanisms by which the constraint of a hard pericarp determines germination timing in response to the environment are currently unknown. Here, we show that the hard pericarp of *Lepidium didymum* controls germination solely by a biomechanical mechanism. Mechanical dormancy is conferred by preventing full phase-II water uptake of the encased non-dormant seed. The lignified endocarp has biomechanically and morphologically distinct regions that serve as predetermined breaking zones. This pericarp-imposed mechanical dormancy is released by the activity of common fungi, which weaken these zones by degrading non-lignified pericarp cells. We propose that the hard pericarp with this biomechanical mechanism contributed to the global distribution of this species in distinct environments.

[1] Department of Biology, Botany, University of Osnabrück, Barbarastraße 11, D-49076 Osnabrück, Germany. [2] School of Biological Sciences, Plant Molecular Science and Centre for Systems and Synthetic Biology, Royal Holloway University of London, Egham, Surrey TW20 0EX, UK. [3] Department of Biology, Center for Advanced Light Microscopy, University of Osnabrück, Barbarastraße 11, D-49076 Osnabrück, Germany. Katja Sperber and Tina Steinbrecher contributed equally to this work. Gerhard Leubner-Metzger and Klaus Mummenhoff jointly supervised this work. Correspondence and requests for materials should be addressed to G.L.-M. (email: Gerhard.Leubner@rhul.ac.uk) or to K.M. (email: Klaus.Mummenhoff@Biologie.Uni-Osnabrueck.de)

A diversity of fruit and seed structures provide biomechanical and ecophysiological adaptations to support reproductive performance and plant fitness in distinct environments[1–5]. 'Hard Seededness' has arisen many times across plant taxons whereby a hard inner layer of the pericarp (fruit coat) encases the seed. The means by which hard endocarps open during germination were first investigated in 1933 by Sir Arthur Hill, Director of the Royal Botanic Gardens (Kew, London). Hill[6] and others[7] found Eocene fossil fruit valves with a hard endocarp. Global climate change processes in the Eocene were identified as the primary selective agents for physical dormancy characterised by water-impermeable hard seed or fruit coats[2, 3, 8]. However, most hard-seeded species have physiological dormancy (PD) with water-permeable seed or fruit coats[9–15]. Seven major dormancy classes, with physical dormancy the most restricted and PD the most common, have been proposed by seed ecologists[3, 16].

Dormancy is an innate seed/fruit property that defines the environmental conditions required for germination[3, 5, 16]. The environmental sensitivity provided by PD appears to be a key trait in the diversification of seed plants. Willis et al.[16], proposed from their phylogenetic analysis, that PD acted as an 'evolutionary hub' from which other dormancy classes evolved. This also includes the non-dormancy (ND) class, which enables seedling establishment as soon as conditions become favourable for germination. Species with ND seeds might be better able to explore novel environments because their germination is independent of specific dormancy-breaking cues that might be absent in that new environment[16]. Diversification and global radiation of the Brassicaceae in the Miocene was associated with the evolution

of distinct seed and fruit traits[10, 12–14, 17–19]. Dispersal of PD or ND seeds from dehiscent fruits is the ancestral condition of the Brassicaceae genus *Lepidium*, comprising ~250 species. The dispersal of indehiscent and didymous-type fruits evolved independently several times within this genus[18, 20–23]. *Lepidium* is therefore highly suited for study of the biomechanics and ecophysiology of pericarp traits in seed protection and dispersal strategies.

Here, we elucidate the biomechanical and ecophysiological mechanisms underpinning the mechanical dormancy imposed by the hard pericarp of *Lepidium didymum* L. (syn: *Coronopus didymus* (L.) Smith) fruit valves. *L. didymum* is a successful weed of South American origin widely distributed over all continents, from the boreal to the tropical climatic zones[18, 23] (Supplementary Fig. 1). It is a problematic weed in agricultural ecosystems including in dairy farming where it causes off-flavoured milk[24] and forms abundant and persistent weed seed banks in arable soil and grassland[25–30]. We report here that the *L. didymum* didymous fruit valve evolved to have a structure that prevents full water uptake of the encased ND seed. This solely mechanical mechanism consists of a hard endocarp with a 'Predetermined Breaking Zone' (PBZ) and a distinct crack initiation zone. To release the pericarp-imposed mechanical dormancy and trigger germination, the PBZ is weakened in its biomechanical properties by the activity of fruit-associated common fungi.

## Results

**Pericarp-imposed mechanical constraint to seed germination.** We found that the hard pericarp of the *L. didymum* fruit valve

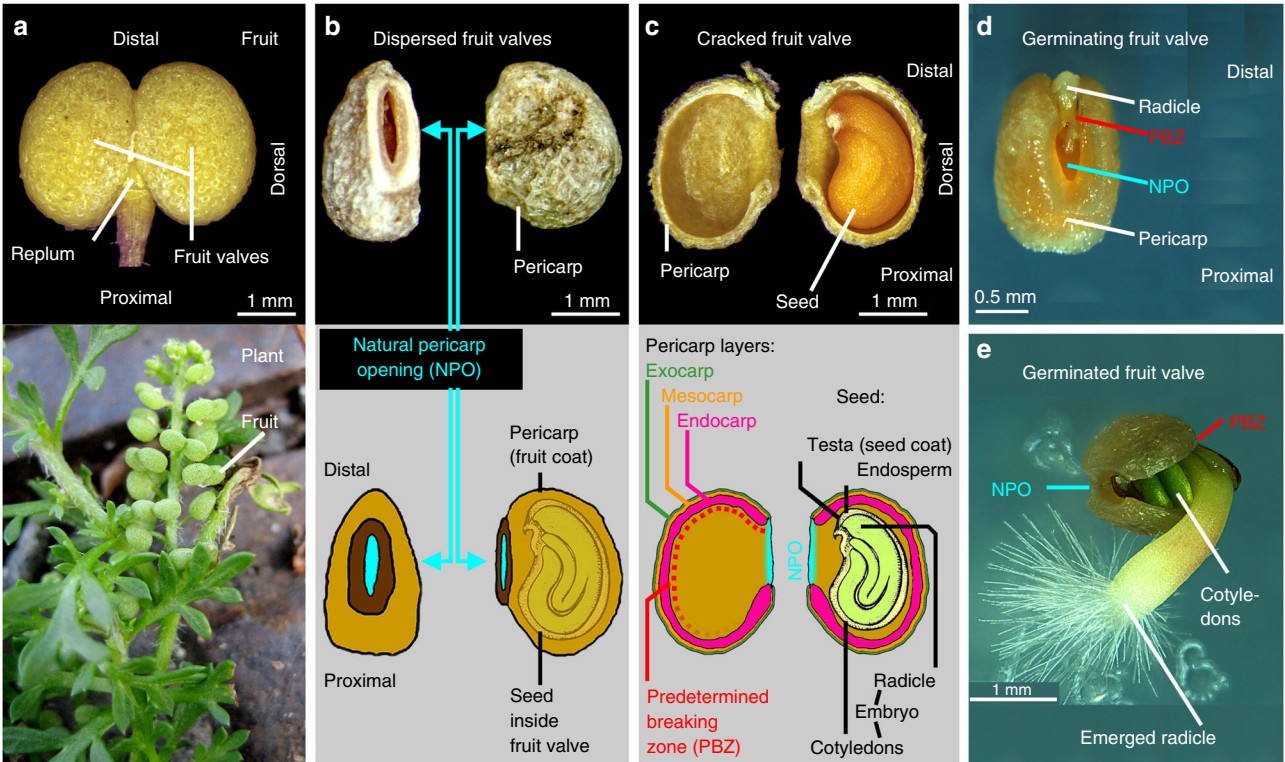

**Fig. 1** The dispersal units of *Lepidium didymum* are fruit valves each harbouring a single seed encased in a hard pericarp (fruit coat). **a** Oblate spheroid-shaped fruits each consisting of two fruit valves attached to the replum. **b** Detachment from the fruit's replum at maturity provides two dispersed fruit valves. The mature fruit valves have a hard pericarp and a 'Natural Pericarp Opening' (NPO) that permits water uptake and gas exchange. **c** Dry fruit valve, cracked open, exposing the encased single dry seed that fills almost the entire pericarp cavity. The seed is always positioned inside the cavity with the radicle (embryonic root) directed towards the distal pericarp region adjacent to the NPO. The pericarp layers and the 'Predetermined Breaking Zone' (PBZ) along the distal and dorsal pericarp regions are indicated. **d** Germinating fruit valve with pericarp rupture initiated in the distal pericarp region (crack initiation point) adjacent to the NPO. **e** Germinated fruit valve opened like a shell by progressed pericarp rupture to facilitate radicle emergence

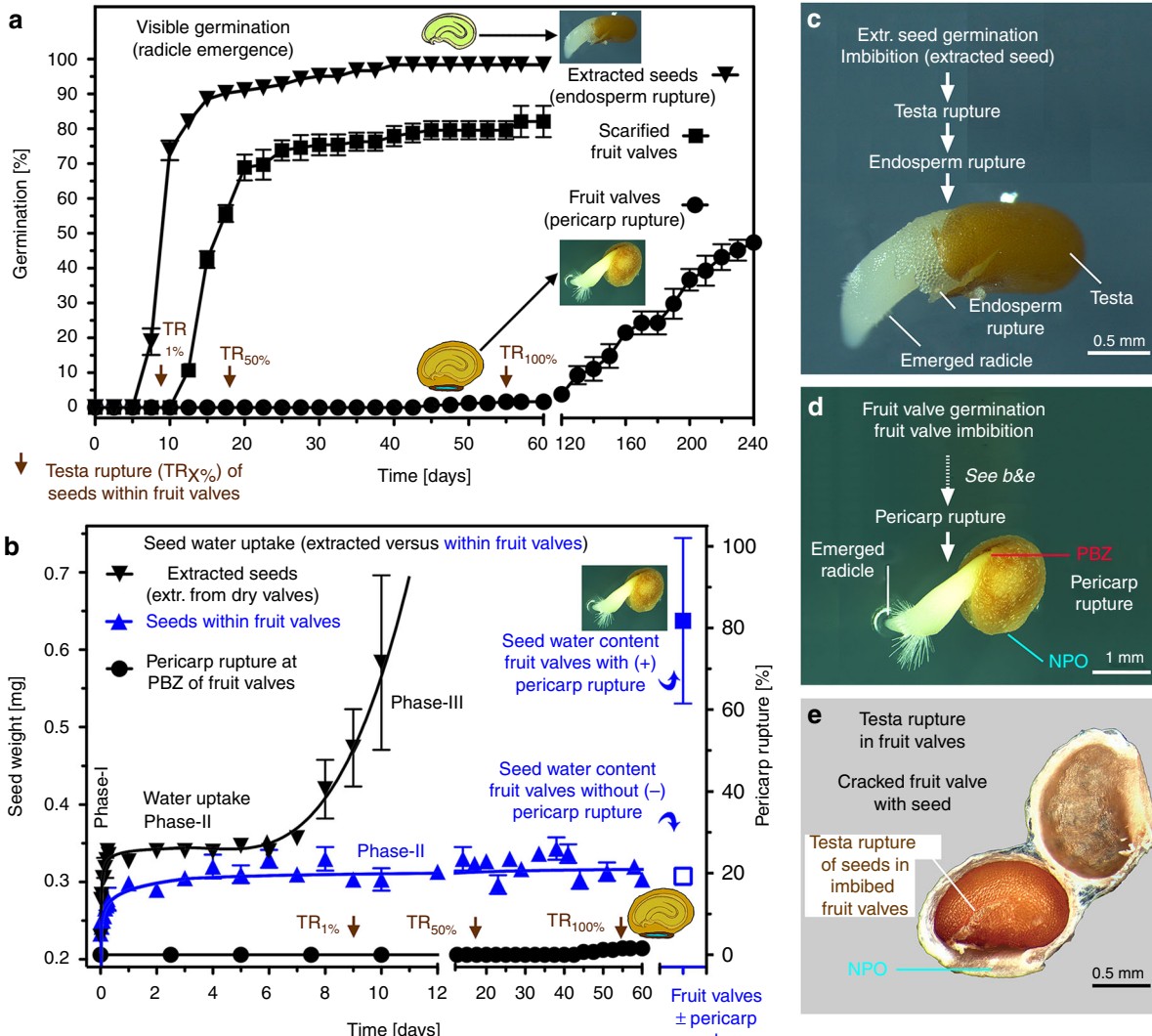

**Fig. 2** The comparative germination of *Lepidium didymum* fruit valves and extracted seeds reveals the mechanical constraint of the pericarp to full phase-II seed water content. **a** Time course of visible germination of surface-sterilised, intact and scarified fruit valves compared to extracted seeds (manually removed from cracked open fruit valves). Note that extracted seeds are non-dormant (ND) and germinate readily in the fresh mature state. Mean values ± SE ($N = 3 \times 50$) of accession KM2423 at conditions identified to be optimal for the germination of fresh, mature seeds (Supplementary Fig. 2a–e) and therefore used as 'standard conditions' (15/5 °C day/night with 12 h photoperiod, white light at ~100 µmol/m²/s[1]). The timing of testa rupture (TR; see **e**) of seeds within fruit valves is indicated (brown arrows) as percentage of the population (onset at $T_{1\%}$ until maximum at $T_{100\%}$). Note that TR is completed prior to the onset of pericarp rupture, and that the TR confirms the ND state of the seeds. **b** Water uptake patterns of extracted seeds compared to seeds within fruit valves. Seed extracted from dry (day 0) fruit valves exhibit a typical three-phasic pattern of water uptake[5]: phase-I (imbibition) is followed by the plateau phase-II (metabolic activation) and upon endosperm rupture the radicle emergence is associated with phase-III water uptake. The water content of seeds within fruit valves without pericarp rupture remained in phase-II. The phase-II water content of seeds within fruit valves was significantly lower compared to the phase-II water content of extracted seeds. Pericarp rupture and radicle emergence were associated with phase-III water uptake of seeds within fruit valves; see Supplementary Fig. 4 for further details and statistical analyses. **c** Extracted seeds germinate with testa rupture followed by endosperm rupture as two visible steps; see Supplementary Fig. 2a–e for a detailed analysis of seed germination. **d** Fruit valves germinate with pericarp rupture as visible step. **e** Fruit valve, cracked open, with seed exhibiting testa rupture, which occurred inside the fruit valve during imbibition. For NPO and PBZ see Fig. 1

imposes a mechanical constraint to the germination of the encased seeds. The dispersal units of *L. didymum* are hard fruit valves with a didymous fruit morphology[18, 23] (Fig. 1). The dispersed fruit valve has a thick, hard pericarp encasing a single seed. Inside the fruit valve cavity, the seed is always oriented so its radicle (embryonic root) end is adjacent to the distal pericarp, while the peripheral end of the seed is adjacent to the proximal pericarp (Fig. 1b, c). The hard pericarp does not completely isolate the seed from the ambient environment due to a 'Natural Pericarp Opening' (NPO); a small hole in the pericarp

surrounded by the scar of the detachment zone (Fig. 1b, c). It is therefore obvious that water uptake by the seed inside the fruit valve is potentially possible via the NPO without the requirement for a change in pericarp permeability, but the NPO is too small to release of the seed from the fruit valve. It takes several weeks until the fruit valve eventually cracks open to facilitate radicle emergence (Fig. 1d, e). The radicle emergence does not occur via the NPO, but is localised to the distal pericarp region adjacent to the NPO. The shell-like fruit valve opening occurs by pericarp rupture, which starts in the distal pericarp (crack initiation zone)

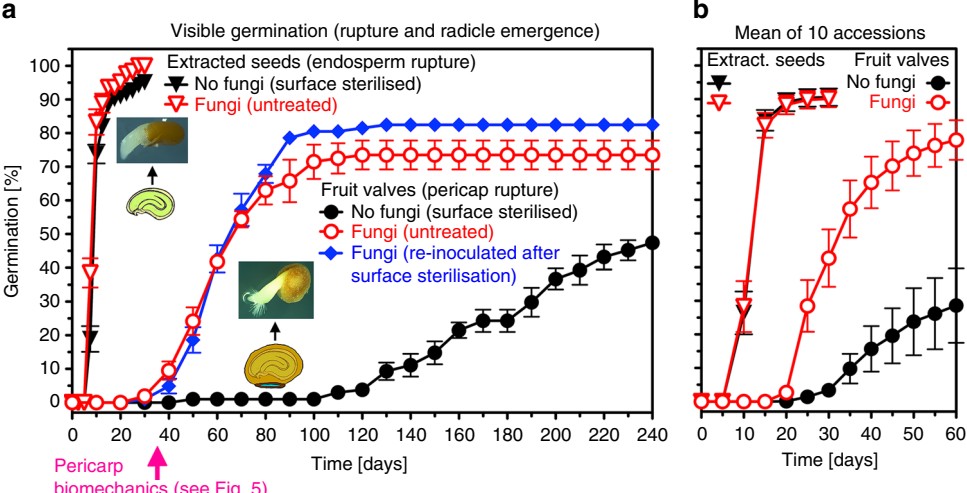

**Fig. 3** The promotional effect of fungal activity on the germination kinetics of *Lepidium didymum* fruit valves. **a** Time course of visible germination of 'untreated' (fungi) compared to 'surface-sterilised' (no fungi) fruit valves. Note that the promotional effect on pericarp rupture was also evident when surface-sterilised fruit valves were re-inoculated with the fungi. The germination kinetics of extracted seeds was not affected by the fungi, nor were the phase-II water content, seed viability or seedling growth (Supplementary Figs. 2 and 4). Mean values ± SE ($N = 3 \times 50$) of accession KM2423 at standard conditions. The time for the pericarp biomechanics (Fig. 5) is indicated (pink arrow). **b** Mean values ± SE of the effect of fungi on the germination of fruit valves and extracted seeds from ten independent *L. didymum* accessions. The individual time courses are presented in Supplementary Fig. 3 and the origin of the ten accessions is listed in Supplementary Table 1

from where it propagates along the distal and dorsal region. Using biomechanical and microscopic analysis, described later, we identified this region as PBZ for pericarp rupture.

Fruit valve germination of *L. didymum* is a slow process spread over weeks or months, but 'extracted seeds' (seeds manually extracted from fruit valves) germinated readily within a few days (Fig. 2a, c, d). To investigate if this pericarp-mediated germination constraint is due to a biomechanical mechanism, we compared these temporal patterns to the germination of scarified fruit valves. Mechanical scarification of fruit valves by pericarp cracking provided artificial pericarp rupture. The germination experiments demonstrated that scarification shifts the onset of visible germination of surface-sterilised fruit valve populations from ~120 days to ~10 days, and the time to reach 50% from ~240 days to ~15 days (Fig. 2a). The similarity in germination timing of the scarified fruit valves and the extracted seeds, as compared to the inhibited germination of intact fruit valves, excludes pericarp-released germination inhibitors (chemical dormancy)[3, 5] as the cause for the pericarp-mediated constraint.

We show here that the pericarp permits partial water uptake by seeds and testa rupture inside fruit valves, but prevents full water uptake required for germination of ND seeds. The completion of germination by radicle protrusion depends on the balance of forces between the growth potential of the embryo (expansion force driven by water uptake) and the weakening of the 'coat' constraints[2, 5]. To further investigate the mechanisms of the *L. didymum* pericarp constraints therefore required that we characterised the dormancy and germination properties of extracted seeds. Their germination occurred with two visible rupture steps; testa rupture, followed by endosperm rupture and radicle emergence (Fig. 2c, Supplementary Fig. 2a–d). This germination pattern is typical of seeds dispersed from the dehiscent fruits of *Lepidium sativum* (ND seeds)[21] and *Lepidium papillosum* (PD seeds)[22]. For the extracted seeds of *L. didymum* accession KM2423 (from Northern Germany) we identified a wide optimal temperature window from 8 to 21 °C and that 15/5 °C (day/night) was optimal for seed germination ('standard condition', Supplementary Fig. 2e). The wide temperature

optimum and preference for cooler temperatures is in agreement with the observed seedling emergence patterns during mild winters in Northern Germany[30] and across seasons in England[26]. The extracted seeds from freshly harvested mature fruit valves of ten independent *L. didymum* accessions (Supplementary Table 1) readily germinated under our standard conditions (Supplementary Fig. 2b). This was well in advance of their fruit valve germination confirming the importance of the pericarp constraint.

To investigate how the hard pericarp imposes a mechanical constraint to the germination of the encased seeds we compared the water uptake by extracted seeds to seeds inside fruit valves (Fig. 2b). This comparison revealed that the hard pericarp is a mechanical constraint and only permits initial (imbibition) and partial phase-II water uptake into seeds inside fruit valves, but prevented the transition to phase-III water uptake required for endosperm rupture and radicle emergence[5]. Interestingly, despite the mechanical constraint preventing full water uptake by the seeds, testa rupture of seeds occurred inside the fruit valves (Fig. 2a, b, e, Supplementary Fig. 4a). This occurred well in advance of the onset of any fruit valve germination, at ~120 days (Fig. 2a). At ~60 days 100% of the seeds inside the fruit valves had testa rupture but none had endosperm rupture (Fig. 2b). Prior to pericarp rupture the water content of seeds within fruit valves remained roughly constant at the low phase-II level, and increased only upon pericarp rupture and radicle emergence (Fig. 2b). Taken together, we conclude that *L. didymum* fruit valves harbour ND seeds and that the hard pericarp imposes a mechanical constraint to the full phase-II water uptake of these ND seeds, delaying germination by pericarp rupture for several months (Fig. 2a).

**Fungi promote pericarp rupture.** In contrast to surface-sterilised fruit valve populations where the onset of pericarp rupture is at ~120 days, untreated fruit valves exhibited a substantially earlier onset at ~35 days (Fig. 3a). Not only the onset, but also the rate of pericarp rupture was enhanced in the untreated fruit valve

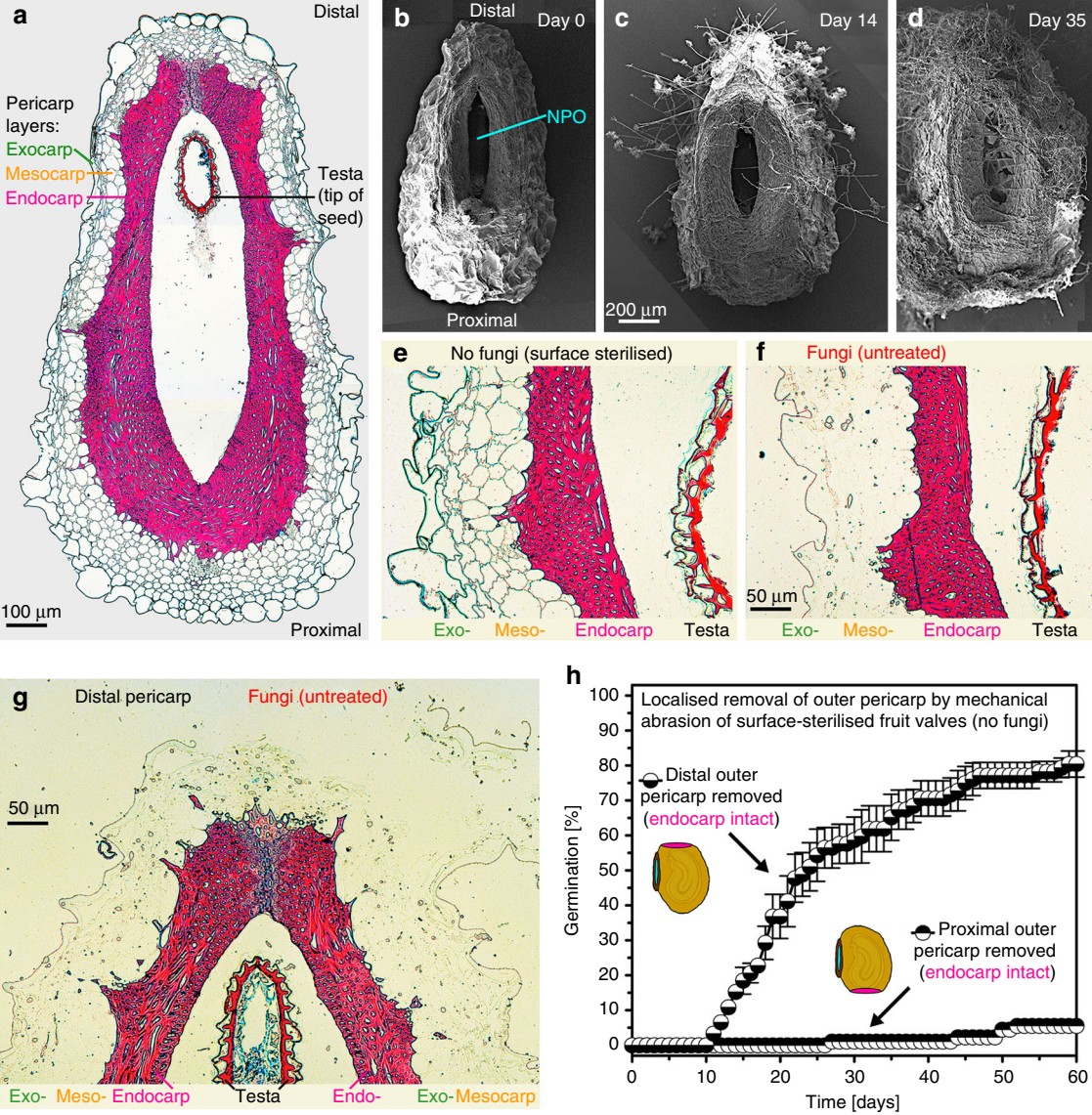

**Fig. 4** Colonisation of *Lepidium didymum* fruit valves with fungi leads to degradation of the outer pericarp layers without visible effects on the lignified endocarp. **a** Light microscopy (LM) of a full longitudinal fruit valve section with safranin-astrablue illustrates the distinct pericarp layers (see Fig. 6a detailing the location of the section). For the endocarp, the intense red staining indicates highly lignified thick secondary cell walls of dead cells. For the exocarp and mesocarp the blue staining of the non-lignified primary cell walls indicates living parenchymatic cells. **b–d** Scanning electron microscopy (SEM) of fruit valves visualising the colonisation of the outer pericarp by fungi during incubation at standard conditions. **e–g** LM of pericarp demonstrating the degradation of the exocarp and mesocarp layers by fungal activity after 7 days of incubation while the lignified endocarp and the seed's testa did not show any visible degradation. The surface-sterilised control (no fungi) shows no pericarp degradation. **h** Effect of localised mechanical abrasion of either distal or proximal outer pericarp on the time course of pericarp rupture of surface-sterilised fruit valves (no fungi). Note that the localised mechanical abrasion was conducted in a way that the endocarp remained intact (see Supplementary Fig. 6 for details)

populations. This reduced the variability in germination timing (4–8 months) to a more uniform germination within 1–2 months. In contrast to the fruit valves, no appreciable effect of the surface sterilisation was evident for the germination kinetics of extracted seeds (Fig. 3a). The finding that the pericarp rupture of untreated, compared to surface-sterilised fruit valves was promoted, while the germination of extracted seeds was not, was evident in ten independent accessions from Europe, Africa and Australia (Fig. 3b; Supplementary Fig. 3).

The *L. didymum* fruit valve pericarp is organised in layers, with the outer pericarp layers (exocarp and mesocarp) covering the inner endocarp layer (Figs. 1c, 4a). The outer pericarp layers comprise living parenchymatic cells with primary cell walls, while the hard endocarp constitutes a layer of dead cells with thick and lignified cell walls (Fig. 4a). Microscopy of untreated fruit valves revealed that the promotion of pericarp rupture (Fig. 3a) was associated with fungal colonisation of the outer pericarp (Fig. 4). Fungal hyphae were first visible after a few days of fruit valve imbibition, and after 14 and 35 days the entire outer pericarp was colonised by fungal hyphae (Fig. 4c, d). This fungal colonisation did not impair germination, seed viability, and the development of healthy seedlings (Supplementary Fig. 2). Re-inoculation of surface-sterilised fruit valves with the fungi also led to the promotion of pericarp rupture (Fig. 3a). The promotion of pericarp rupture by fungi was neither associated with increased phase-II seed water contents inside fruit valves nor with directly enhancing embryo growth, but it triggered earlier pericarp rupture associated with earlier phase-III water uptake and the

completion of fruit valve germination (Supplementary Fig. 4). As this fungal colonisation was also observed on fruit valves harvested directly from the mother plant, we conclude that the promotion of pericarp rupture seems to be achieved by the activity of fruit-associated fungi. To reveal the mechanisms by which the fungal activity releases the pericarp-imposed mechanical constraint we applied microscopic and biomechanical methods.

**Fungi selectively weaken the distal region of the PBZ.** Our finding that the pericarp acts by mechanically preventing full water uptake required for the completion of seed germination (Fig. 2b) suggested that the promotion of fruit valve germination by the fruit-associated fungi (Fig. 3) is achieved by targeted fungal erosion of pericarp tissue to lower its mechanical resistance. To investigate the underpinning biomechanical mechanisms and if

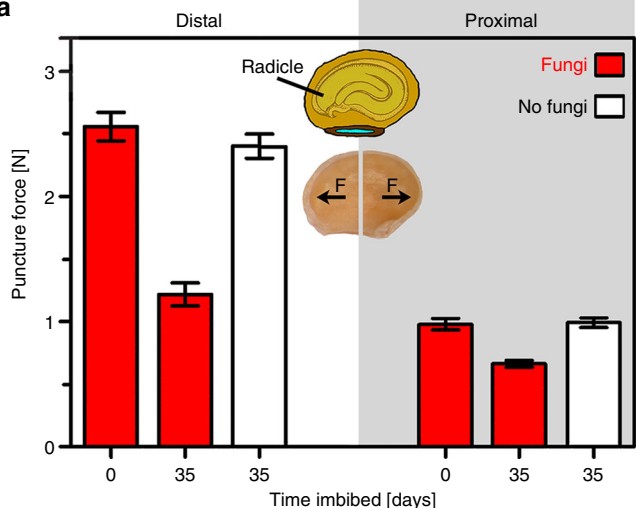

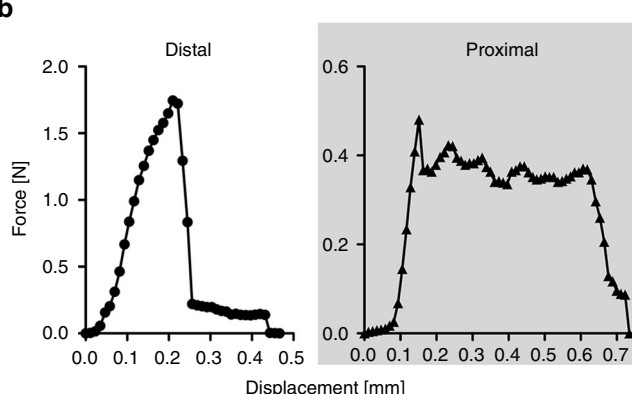

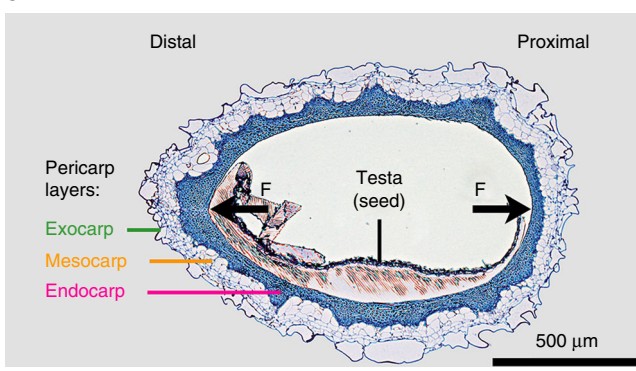

the fungal activity indeed lowered mechanical resistance we performed puncture force measurements on different regions of the fruit valve. Figure 5a shows that the breaking strength of the distal pericarp region dropped drastically from $2.6 \pm 0.1$ N to $1.2 \pm 0.1$ N after 35 days of incubation with fungi. In contrast to this force decrease by ~52% from the initial (day 0) puncture force value, no alteration in the mechanical resistance of the distal pericarp region was evident without fungi (Fig. 5a). The proximal (Fig. 5a) and dorsal (Supplementary Fig. 5) pericarp regions were also affected significantly by the fungi, but to a lesser degree. The force decrease with fungi was only ~30%, with no alteration in the mechanical resistance evident without fungi.

Consistent with a decrease in the mechanical resistance by fungal activity, we found that fungal colonisation indeed eroded the pericarp (Fig. 4e, f). After only 1 week of incubation with fungi, the living, non-lignified exocarp and mesocarp layers were almost completely degraded (Fig. 4f, g), while they remained intact without fungi (Fig. 4a, e). Figure 4f, g also show that neither the testa of the seeds nor the dead lignified endocarp exhibited visible degradation by the fungal activity. To provide direct evidence with a 'no fungi' method that the removal of the outer pericarp layers (exocarp and mesocarp) alone is sufficient for the release of the mechanical pericarp dormancy, we conducted abrasion experiments with surface-sterilised fruits. Figure 4h shows that specific removal of the distal outer pericarp layers by abrasion can fully replace the fungal activity to release the mechanical pericarp dormancy and to promote fruit germination. In contrast to this, specific abrasion of the proximal outer pericarp layers did not lead to dormancy release. In the abrasion experiments where the outer pericarp was removed, the endocarp remained intact (Supplementary Fig. 6). We conclude that the observed degradation of the distal outer pericarp material by fungal activity (Fig. 4g) as well as its specific artificial removal (Fig. 4h) are consistent with the decrease in the mechanical resistance (Fig. 5) and sufficient to promote fruit valve germination (Fig. 3a). The reduction in mechanical resistance

**Fig. 5** Fracture biomechanics of the pericarp and the effect of fungal degradation on the mechanical resistance of distinct *Lepidium didymum* fruit valve regions. **a** Comparative puncture force analysis of the distal and proximal pericarp regions. Fungal activity caused a drastic decrease by ~52% in the mechanical resistance (breaking strength) of the distal pericarp (*p*-value < 0.001) where the pericarp rupture is initiated and the radicle will emerge. A smaller but significant decrease by fungal activity of ~30% was evident in the proximal (*p*-value = 0.04) and dorsal (Supplementary Fig. 2c) pericarp regions. To conduct the biomechanical analyses, fruit valves of accession KM2423 were incubated at standard conditions for the time indicated either untreated (fungi colonise the pericarp) or surface-sterilised (no fungi). No decrease in the mechanical resistance was observed upon surface-sterilisation (distal *p*-value = 0.89; proximal *p*-value > 0.99). Mean values ± SE (N ≥ 26). **b** Comparative force–displacement curves revealing distinct fracture biomechanical properties, namely, sudden complete failure (fatal 'brittle' failure) for the distal pericarp, and slower gradual failure ('composite' failure) for the proximal pericarp. This breaking behaviour clearly identifies this distal region mechanically as the PBZ crack initiation zone (iPBZ) for pericarp rupture, which upon mechanical failure causes the fruit valve to split into half. In contrast to this, in the proximal pericarp the measuring needle was driven through the proximal pericarp, layer by layer until the fruit valve finally split in half. Examples presented are from surface-sterilised fruit valves (35 days); the same difference in breaking behaviours between distal and proximal pericarps were evident for the other conditions. **c** Light microscopy of a full centric longitudinal fruit valve section exhibiting the distinct pericarp layers (toluidinblue histostaining). Arrows indicate force directions for the biomechanical analyses

caused by fungal activity was far more pronounced in the distal, compared to the proximal and dorsal pericarp regions. Together with the finding that abrasion of the distal (and not the proximal) outer pericarp, promotes pericarp rupture, this strongly suggests that a distinct biomechanical structure may cause a specific weakening of the distal PBZ region where the radicle emerges.

**Biomechanical and morphological fruit valve properties**. To investigate spatial differences suitable to explain the mechanisms of pericarp rupture along the PBZ, and to reveal targets for the release of the pericarp-imposed dormancy by fungal activity, we focused on an enhanced biomechanical and microscopy analyses of fruit valve regions. Figure 5b shows the distal and proximal regions of the pericarp have very distinct patterns in their force–displacement curves. At the distal pericarp region, the force increased with displacement up to a maximum value at which a sudden, complete, failure led to a force drop to almost zero. This consistent breaking behaviour (fatal 'brittle' failure)[2, 31] indicates a specialised underlying morphology. Microscopy and biomechanics identifies this distal region as the PBZ crack initiation zone (iPBZ) for pericarp rupture, which, upon mechanical failure causes the fruit valve to split into halves, in the manner observed during germination (Fig. 1d, e). In contrast to the distal region, the proximal pericarp region showed a totally different mechanical behaviour (Fig. 5b). After a critical value was reached, the force dropped gradually in several steps (slow 'composite' failure creating a typical 'zig-zag' pattern)[2, 31]. The distinct curves observed for the two pericarp regions suggest differences in their functional morphology indicating their different roles during pericarp rupture.

The morphological structures and cellular properties underpinning the distinct biomechanical behaviours are shown Fig. 6. Pericarp rupture always starts in the distal pericarp, adjacent to where the radicle is localised. From there it progresses along the PBZ in the distal and dorsal pericarp regions. The pericarp rupture leaves a smooth breaking edge along the PBZ (Fig. 6c–f). In the dorsal PBZ and distal NPO regions rather regular and long endocarp cells are running in parallel (Fig. 6c, e, f). In contrast to this, the proximal NPO region (which does not rupture) is characterised by shorter endocarp cells, organised in a more disordered and interwoven formation (Fig. 6g, h). The longitudinal section of the fruit valve pericarp (Fig. 4a) and the close-up view of the distal pericarp (Fig. 6i) reveal that the distal PBZ endocarp where the pericarp rupture starts (iPBZ) is indeed distinct. The endocarp cells in the distal iPBZ endocarp have a highly regular and parallel orientation, distinct from the adjacent and the proximal endocarp (Figs. 4a, 6i). The distal iPBZ endocarp cells are significantly less lignified compared to the cells of other endocarp regions (Fig. 6 i, k), and its endocarp cell walls contain distinct glycoprotein or hemicellulose epitops (Fig. 6j). In summary, the evolution of the *L. didymum* fruit valve has resulted in morphologically and micromechanically distinct regions as targets for the release of pericarp-imposed dormancy by fungal activity, which decreases mechanical resistance.

**Identification of fruit-associated fungi**. To identify the fungal species promoting pericarp rupture by the degradation of the outer pericarp, we used a universal DNA barcode marker method for distinguishing species of the fungal lineages (Ascomycota, Basidiomycota, early dividing lineages)[32]. Our analysis identified five to six common *Ascomycota* fungi to constitute the *L. didymum* fruit-valve community (Table 1). These ascomycetes, especially *Cladosporium sphaerospermum* and *Aureobasidium pullans*, are among the most common and widespread fungi in

soil and air[33–36]. They are globally distributed in soils of almost all continents and climatic zones (Supplementary Fig. 7). They are also known as seed-borne plant epiphytes and endophytes with proposed roles in seed/fruit traits[15, 37–46]. Seed/fruit microbiomes are composed mainly of *Dothideomycetes*[38], which is also the case for the *L. didymum* fruit valve common ascomycete community (Table 1). Cooperation between the seed-borne microbes is frequent and often leads to protecting germinating seeds and emerging seedlings against abiotic stresses (heat, drought), herbivory and pathogen attack[34, 38, 40, 43, 44]. *C. sphaerospermum* is known as a gibberellin-producing seedling endophyte of soybean[45] and vertical transmission of *C. sphaerospermum* and *Alternaria alternata* via seeds is widespread in herbaceous plants[39]. *A. pullans* is the dominant endophyte of bean seeds and seedlings[41] and an antagonist of pathogenic fungi in post-harvest diseases[34]. Several *Penicillium* species including *Penicillium brevicompactum* and *Penicillium olsonii* were identified as seed-borne endophytes[15, 37, 42, 43], but without revealing the underpinning mechanisms.

Figure 4f shows that the *L. didymum* fruit-valve ascomycetes degraded the outer pericarp without visibly affecting the lignified endocarp. Additional to no lignin degradation, we did not identify any white-rot basidiomycetes. The identified ascomycetes are however known to contribute to the enzymatic degradation of other plant cell wall components. *C. sphaerospermum*, *A. pullulans*, as well as several *Alternaria* and *Penicillium* species produce various cell wall degrading enzymes including cellulases, xylanases, mannanases, arabinases and pectinases[34, 47, 48]. Taken together, this is consistent with our findings that the identified common ascomycetes (Table 1, Supplementary Fig. 7) can degrade exocarp and mesocarp primary cell walls (Fig. 4f) to release the pericarp-imposed mechanical dormancy and promote germination and seedling emergence of *L. didymum*.

## Discussion

The *L. didymum* system revealed that mechanical dormancy imposed by a hard pericarp and its release by the activity of common fungi depend on the fruit valve morphology. The default reproductive strategy within the genus *Lepidium* are dehiscent fruits that open to disperse ND or PD seeds[18, 20–22], but *L. didymum* has evolved didymous-type fruit valves for which we have unravelled the underpinning mechanism. The mechanism by which the hard endocarp inhibits the germination of the encased ND seed is that it imposes a mechanical constraint to full water uptake by the seed required for the completion of germination. Even under our experimental conditions with maximal and continuous water availability, which is not the case in nature, this inhibits fruit valve germination for at least 4–8 months and distributes the germination of the population over many months. The pericarp-imposed mechanical dormancy therefore spreads germination and seedling establishment over and beyond the actual growing season, and explains a key aspect of the observed emergence phenology of *L. didymum*[26, 28]. The pericarp-imposed constraint of *L. didymum* fruit valves to full phase-II water uptake of the encased ND seeds is clearly distinct from seeds were imbibition is completely blocked by water-impermeable seed or fruit coats[3, 8]. Hard endocarps mechanically restricting full water uptake have also been described for (closed) indehiscent fruits with water-permeable pericarps[9–14, 49]. From this work, roles for the hard pericarps were proposed including additional seed protection by reducing hydration and dehydration rates, increasing persistence in soil by retaining seed viability during wet-dry cycles, mechanically enhancing the seed's PD to spread germination and seedling emergence further over an extended period of time (months and years). In neither of these cases were

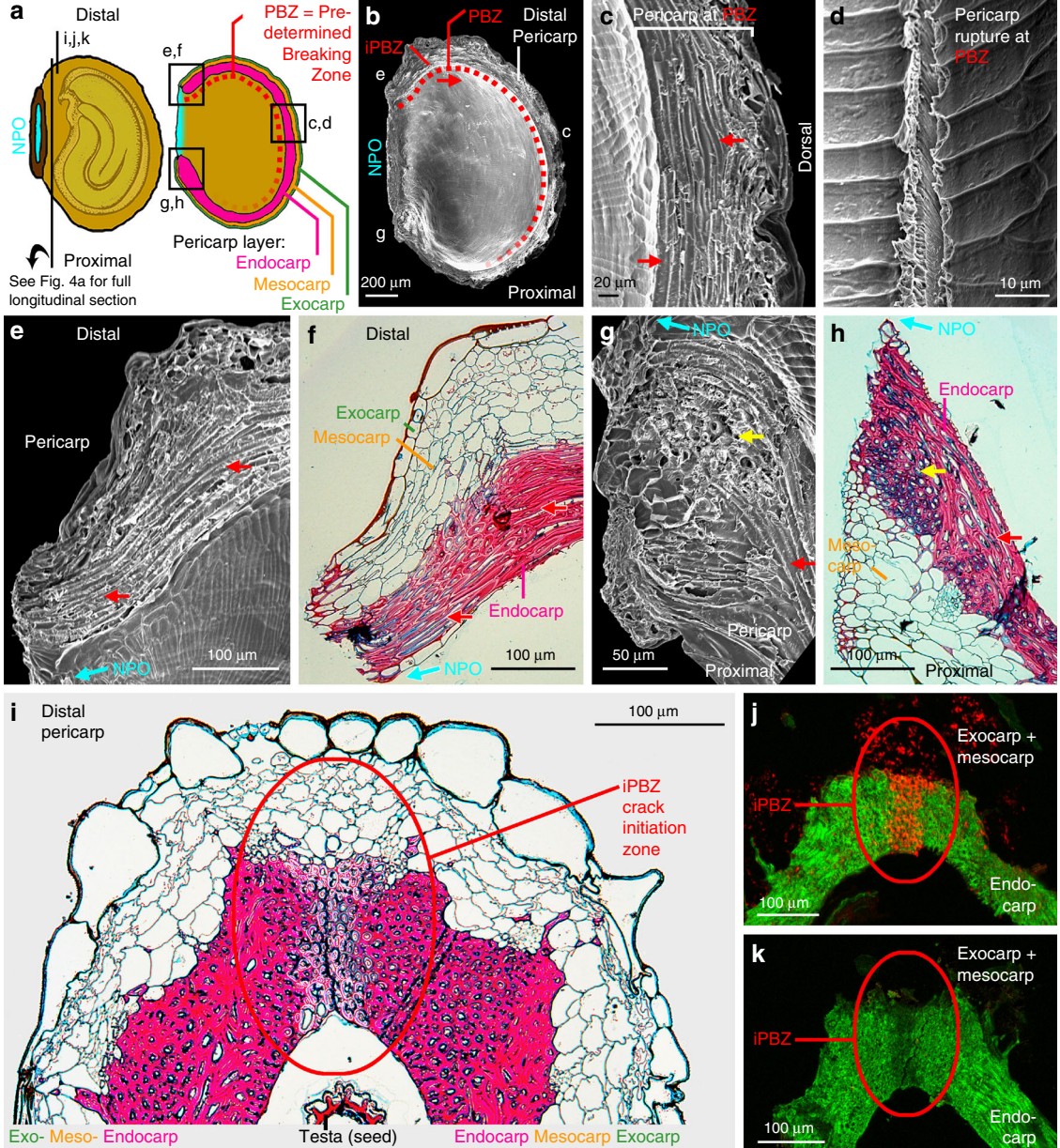

**Fig. 6** Microscopy of the *L. didymum* fruit valve reveals functional–morphological and micromechanical distinct endocarp regions. **a** Schematic drawing of the fruit valve to illustrate the orientation of imaged sections. The seed is always positioned inside the cavity with the radicle (embryonic root) directed towards the distal pericarp region adjacent to the 'Natural Pericarp Opening' (NPO) where the pericarp rupture initiates (iPBZ, crack initiation zone). **b** Scanning electron microscopy (SEM) top view onto the inside wall and the smooth breaking edge along the PBZ. The pericarp rupture spreads from the iPBZ, along the distal and dorsal pericarp as indicated (arrow). **c** SEM top view onto the smooth breaking edge of the lateral pericarp region. **d** SEM view onto the inside valve endocarp at the PBZ. **e** SEM top view onto the pericarp at the distal NPO border. **f** Light microscopy (LM) cross section of the pericarp at the distal NPO border. Red safranin staining indicates intense lignification of the dead thick-walled endocarp cells. Astrablue staining indicates non-lignified primary cell walls of the living parenchymatic exocarp and mesocarp cells. **g** SEM top view onto the pericarp at the proximal NPO border. **h** LM cross section of the pericarp at the proximal NPO border. **i** LM longitudinal section (safranin-astrablue histostain) of the distal pericarp with the iPBZ (crack initiation zone). Reduced lignification of the iPBZ endocarp compared to the adjacent endocarp is evident. **j, k** Fluorescence microscopy of the distal pericarp with the iPBZ. **j** Red fluorescence in the iPBZ due to binding of wheat germ agglutinin (WGA, conjugated with Alexa Fluor 633 nm) indicates distinct glycoprotein or hemicellulose composition of the iPBZ endocarp compared to the adjacent endocarp. **k** Autofluorescence (control without WGA) supports reduced lignification of the iPBZ. **c**, **e–h**, Red arrows, parallel oriented long endocarp cells. Yellow arrows, cross-sectioned endocarp cells

the underpinning biomechanical mechanisms of the pericarp-imposed constraints and their release studied.

The mechanism by which the pericarp-mediated mechanical dormancy is released by the activity of common ascomycetes to promote germination is based on the adaptive biomechanical and morphological properties of the *L. didymum* fruit valve. The outer pericarp layers consist of living parenchymatic cells with primary cell walls, while the hard endocarp is a dead layer consisting of cells with thick and heavily lignified secondary cell walls. We identified regions within the pericarp with distinct properties consistent with their roles and mechanisms during dormancy release and germination. We identified the PBZ in the distal and dorsal pericarp, and the crack initiation zone (iPBZ) in the distal pericarp region adjacent to the seed's radicle end. Pericarp

**Table 1 Community of common fungi on *Lepidium didymum* fruit valves**

| Species[a,b] | Family | Class | Division |
|---|---|---|---|
| *Cladosporium sphaerospermum*[c,S] | Davidiellaceae | Dothideomycetes | Ascomycota |
| *Aureobasidium pullulans*[S] | Dothioraceae | Dothideomycetes | Ascomycota |
| *Alternaria alternata*[d,P] | Pleosporaceae | Dothideomycetes | Ascomycota |
| *Alternaria tenuissima*[d,S] | Pleosporaceae | Dothideomycetes | Ascomycota |
| *Penicillium brevicompactum*[S] | Trichocomaceae | Eurotiomycetes | Ascomycota |
| *Penicillium olsonii*[S] | Trichocomaceae | Eurotiomycetes | Ascomycota |

Fungi were identified by using universal DNA barcode marker for species of the distinct fungal lineages (*Ascomycota, Basidiomycota*, early dividing lineages)[32]
[a] All nucleotide blast hits were characterised by a E-values<$3^{-113}$ and 99–100% identity over 98–100% of the sequenced 0.42–0.55 kb fungal nuclear ribosomal internal transcribed spacer (ITS) region DNA fragments (20.12.2016). The identified ITS sequence GenBank accession numbers are MG252479, MG252478, MG252477, MG252480, MG252481
[b] Commonly observed life style: saprophyte (S), pathogen (P)
[c] Highest abundance among the identified fruit valve fungi
[d] Equal nucleotide blast result values and therefore not possible to distinguish

rupture is initiated at the iPBZ and the PBZ guides its propagation in an 'opening fracture mode'[31] along the distal and dorsal pericarp. This role is supported by the parallel orientation of their thick-walled, lignified, endocarp cells and its biomechanical fracture behaviour by fatal 'brittle' failure[2, 31] (Fig. 5). The notch-like shape of the distal endocarp results in higher localised stress upon embryo expansion. In contrast, the proximal pericarp region adjacent to the round cotyledon end of the seed is distinct in shape and does not play a role as part of the PBZ. It also has a more disordered and interwoven pattern of its thick-walled, lignified endocarp cells and a slower 'composite' failure breaking behaviour. A predetermined breaking point as an adaptation to a dynamic environment has also been identified in the basal stem section of aquatic plants[50]. Morphological and biomechanical properties of plant dispersal units constitute fundamental traits of adaptive value to ensure survival in response to prevailing abiotic and biotic environmental factors[1, 3, 4].

Considering the relevance of this fruit fracture biomechanics and the global activity of common ascomycetes in a germination ecological context is important for such a system. For the cosmopolitan weed *L. didymum*, pericarp-imposed mechanical dormancy determines the timing of fruit valve germination and therefore any subsequent life history success including for colonisation, establishment and reproduction. Any abiotic (soil moisture and temperature) and biotic (common ascomycete activity) environmental factor must therefore be present and operate appropriately to explain the observed germination responses and emergence patterns of *L. didymum* in the diversity of its climates and habitats. The world-wide distribution of *L. didymum* (Supplementary Fig. 1) coincides with the world-wide distribution of the identified common ascomycetes (Supplementary Fig. 7). The identified fruit-valve ascomycetes, especially *C. sphaerospermum* and *A. pullans*, are indeed among the most common and widespread fungi[33–36], and are known as seed-borne epiphytes and endophytes[15, 37–44, 46], and are in general mutualists or commensalists in soil and are associated with a wide range of plants.

For the identified common ascomycetes to serve as a biotic environmental factor, their pericarp-degrading activity should relate to the operating post-dispersal germination strategy and observed seedling emergence patterns. A soil seed bank persistence of at 5–10 years was determined for *L. didymum* fruit valves[26, 28], and in a burial experiment with 70 species it ranked among the top three for appreciable numbers of seedlings emerging in the fourth and fifth year[26, 28]. The germination strategy of *L. didymum* is characterised by spreading seedling emergence over the entire growing season with a tendency in spring and autumn[24–26]. The observed seasonal dynamics and soil moisture dependence of fungal communities[34, 36] would resultant in patterns in fungal activity (cell-wall degrading enzymes), which support such a temporal spread of *L. didymum* seedling emergence. Further to this, *L. didymum* is a pioneer species that reacts within weeks, with seedling emergence, to mechanical soil disturbance (e.g. gap formation, crop planting)[27, 29, 30]. The fast seedling flush response upon mechanical soil disturbance[27, 29, 30] connects well with work on different tillage management practices on soil fungal communities in agroecosystems[33]. Tillage is known to increase the abundance of *Cladosporium*, *Penicillium* and *Aureobasidium* and other ascomycetes in soil[33] and the increased fungal activity by the mechanical soil disturbance may stimulate flushes of *L. didymum* seedling emergence. Further ecological work is of course needed to directly link spatiotemporal patterns of *L. didymum* emergence with fruit valve colonisation and pericarp degradation by common ascomycetes. We propose that *L. didymum* exploits the responsiveness of common ascomycetes to local abiotic cues in order to time the release of the pericarp-imposed mechanical dormancy. Outsourcing of this function to ubiquitous fungi may explain *L. didymum*'s own cosmopolitan distribution.

## Methods

**Plant materials and growth conditions**. If not otherwise indicated, experiments were conducted with *L. didymum* accession KM2423 collected in Osnabrück, Germany. The origin and collection information for the 10 studied accessions is given in Supplementary Table 1. Plants were grown in the greenhouse and freshly harvested mature fruit valves were collected from dry infructescences. They were dried in a desiccator with silica gel for 7 days and stored in airtight containers at −20 °C until required.

**Germination experiments**. For germination assays, intact fruit valves or 'extracted seeds' (botanically 'true' seeds manually extracted from fruit valves) were placed onto 1% (w/v) agar plates (9 cm petri dishes) with 0.043% (w/v) Murashige and Skoog basal salt medium (Duchefa, Haarlem, the Netherlands) pH 7.0, referred to hereafter as rooting medium (RM). For germination assays with surface-sterilised seeds or fruit valves 0.1% (v/v) PPM (Plant Preservative Mixture, Plant Cell Technology, Washington, USA) was added to prevent microbial growth. Petri dishes were sealed with parafilm and were incubated at 15/5 °C day/night temperatures with a 12 h photoperiod, white light at ~ 100 µmol/m²/s¹ (standard conditions). Germination was scored over time as visible radicle protrusion using a binocular microscope. Mean values ± SE were calculated from triplicate plates (*N* = 3 × 50). Mechanical scarification of fruit valves by pericarp cracking (using forceps) provided artificial pericarp rupture (Supplementary Fig. 5b). Localised mechanical abrasion of the outer pericarp was as described in Supplementary Fig. 6. Surface-sterilisation of fruit valves or extracted seeds was achieved by incubation in 1% (v/v) NaOCl, 0.02% (v/v) Tween 20 for 1 min, followed by four times for 2 min rinsing with sterile water. Re-inoculation with fungi of surface-sterilised fruit valves (Fig. 3a) was with fungal solution obtained from KM2423 fruit valves incubated for 5 days at 24 °C in darkness on 1.5% (w/v) agar plates with 2% (w/v) malt extract, pH 6.5, 0.06% (w/v) chloramphenicol.

**Universal DNA barcoding of fungi**. Nuclear internal transcribed spacer (ITS) barcoding was used to identify fruit valve associated fungi. Small pieces of the fungal mycelium colonising *L. didymum* KM2423 fruit valves were transferred to 2% (w/v) malt extract, 0.6 µl/ml chloramphenicol, 1.5% (w/v) agar plates and incubated at 24 °C for 3 days in darkness. Colonies were then re-isolated and

regrown on agar plates. Isolated fungal colonies were distinguished from each other by their characteristic features (e.g. growth and colour). Fungal colonies were transferred to 150 ml malt extract liquid culture without chloramphenicol and incubated at 28 °C for 4 days (100 U/min). Fungal cultures were filtered through Miracloth and ground in liquid nitrogen with mortar and pestle. Fungal DNA was extracted and purified as described[51]. DNA of the fungal ITS regions was PCR amplified by 40 cycles (1 min 94 °C, 45 s 50 °C, 2 min 72 °C) using different ITS primer combinations (18F-25R; 18F-5R; or 5F-25R)[52]. Amplification products were purified using the Nucleospin Gel and PCR Clean-up purification kit (Macherey-Nagel, Düren, Germany), ligated into plasmid vector pGEM-T (Promega, Mannheim, Germany) and transformed into E. coli strain DH5α. Plasmid DNA containing inserts were isolated from bacterial colonies and the PCR products were sequenced on an ABI Prism 377 automated sequencer with dye terminator chemistry (Life Technologies, Darmstadt, Germany). Sequences were uploaded to the NCBI BLAST nucleotide database to test for similarity/homology.

**Scanning electron microscopy.** Fruit valves were used intact (Fig. 4) or opened artificially to study the anatomy of the PBZ (Fig. 6). For top views onto the PBZ at the interior side of the fruit valve, fruit valves were cut open with a razor blade. Before observation, samples were mounted on specimen stubs using a carbon adhesive disc (Plano, Wetzlar, Germany), then dried for 2 weeks in a desiccator, and coated with platinum-iridium with a sputter coater (K575X Turbo, Quorum Technologies LTD, Kent, UK). Fruit samples were analysed with a scanning electron microscope (Supra 55VP, Carl Zeiss, Oberkochen, Germany).

**Sectioning and light microscopy.** Fruit valves were fixed overnight at 4 °C in 2.5% (v/v) glutaraldehyde, buffered with 1× PBS (137 mM NaCl, 2.7 mM KCl, 10 mM $Na_2HPO_4$, 2 mM $KH_2PO_4$, pH 7.4). After rinsing in PBS (5×), samples were dehydrated in a graded ethanol series followed by 10 min incubation in a 1:1 mixture of ethanol (99.8%) and propylene oxide and 10 min propylene oxide. Samples were infiltrated in a 1:3 mixture of the embedding medium (Epon resin, Serva Electrophoresis, Heidelberg, Germany) with propylene oxide overnight. Subsequently, samples were transferred into fresh embedding medium (3×) for 5 min at 60 °C. Samples were transferred into embedding moulds and polymerisation was carried out (60 °C, 48–72 h). This procedure follows the methodology described by Purschke and Nowak[53]. Semi-thin sections (0.5–0.7 μm) were cut using an ultramicrotome (Ultracut E, Reichert Microscope Services, Depew, USA) with a diamond knife (Diatome histo jumbo 6.0 mm, Science Services, München, Germany). The embedding medium was removed as described[54] prior to staining with safranin-astrablue histostain.

**Wheat germ agglutinin-Alexa Fluor 633 staining.** Fruit valve samples were embedded in Pattex two-component adhesive (Henkel, Düsseldorf, Germany). After drying for 2 h, samples were cut in 25 μm sections by using a rotary microtome (Leica RM2125RT, Wetzlar, Germany). Sections (Fig. 6j) were stained with 1% (w/v) WGA-Alexa Fluor 633 (Invitrogen, Paisley, UK) as described[55], while controls (Fig. 6k) remain unstained. Sections on microscope slides were washed twice with Hank's balanced salt solution (Invitrogen, Paisley, UK) and subsequently transferred to distilled water. Confocal laser scanning microscopy (FluoView 1000, Olympus, Tokyo) was conducted (Excitation: 635 nm, Emission 650–750 nm) with imaging of z-stacks performed with a ×20 air objective (UPLSAPO, NA 0.75, Olympus) at a voxel size of $0.621 \times 0.621 \times 5$ μm. Lignin auto-fluorescence was detected by laser excitation at 405 nm with a distinct emission peak at ~450 nm (emission bandpass 420–470 nm). Both signals, WGA-Alexa Fluor 633 and lignin, were separated by two channel sequential acquisition. For analysis, maximum intensity projections were calculated from z-stacks and background subtraction with subsequent contrast enhancement were performed (FIJI[56], NIH ImageJ[57] distribution). It is known that WGA binds to specific glycoprotein[58] or hemicellulose[59] epitopes in plant secondary cell walls.

**Biomechanical measurements.** Puncture force experiments were carried out on dry fruit valves of accession KM2423 using a custom-built material testing machine[2, 60] using a 20 N load cell. A rounded measuring tip (Ø 0.5 mm) was lowered to the sample at a constant speed (0.7 mm/min), while force and displacement were recorded simultaneously. To study the influence of fungal colonisation on the material properties of the pericarp samples, fruit valves with (untreated) and without (surface sterilised) fungal colonisation were compared at 0 and after 35 days of incubation at standard conditions. Fruit valves were cut in half using a razor blade. The seed was removed from the pericarp and the empty fruit valves were placed on a sample holder (cylindrical metal sample holder with a 1.5 mm hole). Distal, proximal and dorsal parts of the fruit valves were tested (N= 26–32). One way ANOVA was carried out followed by Tukey's multiple comparisons test using GraphPad Prism 7 (GraphPad Software Inc., CA, USA, version 7.02).

**Data availability.** Fungal ITS sequences were submitted to GenBank and their accession numbers were included in Table 1. The numeric raw data of this article was uploaded to the Figshare repository (doi:10.6084/m9.figshare.5532412).

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

## Acknowledgements

We thank N. Scarlett, J.F. Castro, D. Zacharias, W. Grenzheuser for *L. didymum* seed samples, U. Coja for discussion and help with fungal identification, F. Przesdzink for technical assistance, G. Purschke and Hans-Peter Schmitz for critical discussions, and the Botanical Garden Osnabrück for help with seed propagation. We acknowledge the following agencies who support our work on *Lepidium* seed dormancy: Deutsche Forschungsgemeinschaft (DFG) to K.M. (MU137/8-2), and Biotechnology and Biological Sciences Research Council (BBSRC) to G.L.-M. and T.S. (BB/M000583/1). The support for the seed-microbe project by Azotic Technology to G.L.-M. and J.E.H. is gratefully acknowledged.

## Author contributions

K.S., T.S., K.G., G.L.-M. and K.M. planned and designed the research; K.S., T.S., N.W., S.C. and G.S. performed experiments; K.M. and T.S. provided material or analytical tools; K.S., T.S., J.E.H., R.K., K.G., K.M. and G.L.-M. analysed and interpreted the data; G.L.-M., K.M., T.S., K.G. and J.E.H. wrote the manuscript.

## Additional information

**Competing interests:** The authors declare no competing financial interests.

