## [Peer Review File · Nature Communications]

Reviewers' comments:

Reviewer #1 (Remarks to the Author):

This is an interesting paper which implicates the biological activity of symbiotic fungi in fruit dehiscence. The essence of the findings are that the hard pericarp surrounding seeds of many species of the genus *Lepidium* is contingent on the enzymatic degradation of the fruit cell walls for germination of the seed through the fruit tissues to commence. To the most part the conclusions of the manuscript are supported by the data presented (but see below). This is, to my knowledge, the first study which shows that fungal activity can be important for seed dispersal, and adds an interesting twist on a trend in the seed literature to focus more on the maternal aspects of the control of seed dormancy, germination and dispersal behaviour. In my view there will likely be wide general interest in the new aspect of plant-fungi symbiotic behaviour. I have a few comments that could be addressed by the authors to further improve the manuscript.

Major point:

1. In figure 6 it is proposed that there is a 'crack initiation point' at which pericarp rupture occurs. This evolutionarily makes sense because the zone resembles the dehiscence zone in dehiscent fruits of the Brassica family, such as *Arabidopsis*. However, the results of the fungal activity documented in Figure 4f seem to indicate a general degradation of the exo- and mesocarp, and no effect is shown on the lignified endocarp layer and especially the author-defined 'crack initiation zone'. In this regard it remains unclear exactly how the fungi promote rupture at the 'crack initiation zone'. At a minimum this should be discussed, but ideally further micrographs would show how or whether fungal activity affects the tissues that actually appear to matter in the germination process. In essence it is important to define more precisely how the presence of fungi leads to rupture at the crack initiation zone.

Alternatively it could be further considered whether the exo- or mesocarp limit water uptake into the seed, and thus prevent embryo tissue expansion required for crack initiation. In this scenario the microscopy in Figure 6 becomes more incident to the story, given that the presence of these lignified zones is well documented in the Brassicaceae. The data in figure 2 appear to suggest that water uptake is not important but I was slightly unclear because of point 2 below. After fungal activity, how is pericarp permeability affected?

Minor points:

2. Figure 2b is hard to understand: I wonder if it can be separated to ease comprehension of the data presented. The relevance of the water uptake to the causation of germination could be discussed more clearly. Also, in Figure 3b the symbol labels are only partly clear.
3. In the discussion the potential roles of individual fungal species are discussed, but in the data presented no attempt is made to distinguish between the activities of the different fungal species. Is this discussion a little speculative without further data?
4. The proliferation of acronyms in the manuscript may hinder comprehension by a more general audience.
5. The last two paragraphs (page 10) of the results section strays into discussion. The discussion raises obvious question is whether the fungi can be added back individually to show which are important in the fruit degradation?
6. The manuscript title could be modified to be more informative in respect to the conclusions.

Reviewer #2 (Remarks to the Author):

The manuscript by Sperber et al. gives a detailed description on the mechanisms of seed germination

in *Lepidium didymium* which species is characterized by seeds covered with impermeable valve (fruit) tissue.

The results show convincingly the involvement of fungi to reduced the permeability of the surrounding valve tissue and provides detailed information on the organization of the latter tissue linked to biomechanical analysis.

Although the data are convincing, one wonders what is the novelty of this paper because several aspects of this mechanism have been described before (see cited literature). It would be useful that also in the introduction it is clearly stated what questions were left for the scientific issue addressed here. Now these aspects sometimes only came back in the discussion.

The paper in general is well writing although this reviewer was a bit surprised by the sentence on page 3 about greenhouse warming. It seems that sometimes people like to have some of this modern items in all papers about ecology. The legend of fig 4 is a bit confusing. It tells that a-d are SEM pictures but this is only the case for Fig 4 b-d.

Reviewer #3 (Remarks to the Author):

This is a thorough investigation of the physical control of germination by a hard multilayered pericarp in *Lepidium didymum*, a widely distributed species of *Lepidium*. I would prefer terminology like widely distributed over cosmopolitan which means: familiar with and at ease in many different countries and cultures. I have no major complaints about the research or its interpretations. I think the paper could be condensed slightly without a loss of meaning.

I would prefer to see paragraphs indented when the block style is not used. This makes it much easier for the reader to determine where new paragraphs begin.

Reviewer #4 (Remarks to the Author):

I found this manuscript to be well-written and informative. It provides novel information that I feel will be of interest to persons in the field as well as those generally interested in the soil/seed microbiome. The manuscript contains a significant amount of work and I found the experiments to be properly designed and analyzed. The conclusions are generally supported by the data and the manuscript is appropriately supported with referenced material. I have a few suggestions for the authors.

Summary – the summary does not contain any specific supporting data for conclusions stated. The summary could benefit from inclusion of some specific research data, like the percentage decrease in puncture force related to fungal association, etc.

Introduction- I would prefer to see a set of well-defined objectives at the end of the introduction rather than a pre-conclusion statement.

Materials and Methods – The puncture force description indicates that a mean separation (Tukey's test) was conducted, but only SE is reported on Fig. 5a. Even though the p-value is reported in the figure legend, I would suggest replacing the error bar with letter-represented mean separation.

I think scarification should be described in a single sentence in the manuscript rather than in supplemental information. It is important to understanding data presented in several figures and there are so many ways researchers approach scarification that I think it should be described. Also, although the treatment presented here falls under the broad heading of scarification, I think pericarp cracking would be a more descriptive term to indicate not only the treatments potential impact of

water movement into the fruit, solute egress from the fruit, but also the dramatic reduction in mechanical resistance.

Results and discussion –

I do not feel that the authors have fully supported their conclusion that the pericarp acts to restrict imbibition. I think that it is equally possible that the pericarp is restricting water uptake associated with post-lag events. I think this is not just semantics but is fundamental to the conclusion being drawn. Unless, the authors have reference points to the end of imbibition and initiation of lag period, it is not possible to differentiate between the pericarp restricting imbibition vs. water uptake prior to germination. Given the data provided, I think using the term water uptake is the more conservative approach because it is more inclusive.

The authors also indicate that the pericarp through solely biomechanical restraint is responsible for cessation of water uptake. However, given the fact that seeds eventually increase in fresh weight and germinate (under aseptic conditions), there could be a slow gradual change in embryo growth potential (possibly by after-ripening) that impacts gradual water uptake? Embryo growth potential measurements over the 2 to 4 months required for aseptic germination would indicate whether the embryo was involved in this phenomenon.

I agree with the authors that fungal activity is associated with a reduction in mechanical restraint in the pericarp. However, I am not sure they have provided definitive evidence that the portion of the pericarp responsible for imposing dormancy is due only to the parenchyma-like, thin walled meso and exocarp cells and not the endocarp cells associated with the distal portion of the fruit. I agree with the authors, that the outer pericarp is the most likely target for fungi-associated degradation as shown in Figure 4e, but I would have liked to see a non-fungal outer pericarp removal experiment to provide evidence that removal of these two tissues alone was sufficient for dormancy release. If it was not possible to physically separate these tissues, then I think it should be stated in the manuscript. Also, pericarp sections for fruits at 0 and 35 days with and without fungal association to show that only the exo and mesocarp cells were impacted and not the distal section of the endocarp would provide further support for their conclusion. Figure 4e does show a general lack of cellular change in endocarp cells, but I was particularly interested in any localized changes in the cells over the radicle indicated as iPBZ in Figure 6. It is quite plausible that these initial break cells are also involved in mechanical restraint and could be locally impacted by fungal enzymes. Did the authors take sections in the distal region similar to those in Fig. 4e that could provide additional information? These previous comments are not meant to be critical of the conclusion that fungal activity is associated with reduction of pericarp mechanical restraint, only the specific localization of that impact on the out portion of the pericarp alone.

The discussion is mostly a restatement of results and I think it could be easily condensed.

Reviewer #5 (Remarks to the Author):

This paper presents the results of detailed studies on the biomechanical and ecophysiological characteristics of seed morphology and germination behavior of *Lepidium*, which exhibits mechanical dormancy where the hard pericarp prevents seeds from fully imbibing water from the surrounding environment. A novel finding is that fungi appear to play a key role in degrading targeted pericarp tissues which allows full imbibition and germination to occur. A characteristic group of seed-inhabiting fungi were identified and shown to be capable of degrading pericarp tissues. This conclusion was aided by comparisons of germination of seed with the pericarp removed (eliminating biomechanical constraints on imbibition and germination) and by surface sterilizing seeds to prevent fungal colonization of pericarp structures. In addition, detailed biomechanical and microscopic studies showed

that specific regions of the pericarp differ in such a manner that a crack or rupture point originates in a very specific region, consistent with the idea of adaptive morphology enhancing seed germination, and facilitated by fungal degradation.

Overall, I found this to be an interesting and well written paper with excellent imagery. Fig. 3A especially shows the interaction of fungi with pericarp-associated dormancy. However, I question the level of general interest in this paper in a broad-based journal like Nature Communications compared to a more specialized botanical journal where the detailed morphological and biomechanical studies might be more highly appreciated. While the role of fungi in the germination process is novel, it is unclear whether most other soil fungi could accomplish the same pericarp degradation or whether the five identified taxa are consistently associated with *Lepidium* across its worldwide distribution. Thus, while the paper and study are very well done, it is not clear to me whether the paper would be of interest beyond a small group of specialists.

Two specific comments include:

Discussion – *Lepidium* has a worldwide distribution so it occurs in many climatic conditions and great variation in precipitation, temperature, etc. where fungal activity would also differ. So it was unclear to me how this one mechanism of breaking dormancy could operate successfully over such a wide range of conditions.

Table 1 – The table lists common fungi but what about other fungi? What fraction of OTUs did these five taxa represent? The text says 5-6 common fungi but little detail is provided. Would any type of generic soil fungi have the same activity on pericarp degradation vs. a group of highly coevolved, seed-associated fungi?

Note that in the revised manuscript all changes are in blue.

Reviewers' comments and **our response (in bold)**:

Reviewer #1 (Remarks to the Author):

This is an interesting paper which implicates the biological activity of symbiotic fungi in fruit dehiscence. The essence of the findings are that the hard pericarp surrounding seeds of many species of the genus *Lepidium* is contingent on the enzymatic degradation of the fruit cell walls for germination of the seed through the fruit tissues to commence. To the most part the conclusions of the manuscript are supported by the data presented (but see below). This is, to my knowledge, the first study which shows that fungal activity can be important for seed dispersal, and adds an interesting twist on a trend in the seed literature to focus more on the maternal aspects of the control of seed dormancy, germination and dispersal behaviour. In my view there will likely be wide general interest in the new aspect of plant-fungi symbiotic behaviour.

We are thankful for this comment and our works will indeed be of wide general interest and trigger a paradigm change and discussion beyond fields. Both, the seed-fungi interaction and the underpinning biomechanics are novel mechanisms. In the revised version we have included results from additional experiments which further support our conclusions.

I have a few comments that could be addressed by the authors to further improve the manuscript.

Major point:

1. In figure 6 it is proposed that that there is a 'crack initiation point' at which pericarp rupture occurs. This evolutionarily makes sense because the zone resembles the dehiscence zone in dehiscent fruits of the Brassica family, such as *Arabidopsis*. However, the results of the fungal activity documented in Figure 4f seem to indicate a general degradation of the exo- and mesocarp, and no effect is shown on the lignified endocarp layer and especially the author-defined 'crack initiation zone'. In this regard it remains unclear exactly how the fungi promote rupture at the 'crack initiation zone'. At a minimum this should be discussed, but ideally further micrographs would show how or whether fungal activity affects the tissues that actually appear to matter in the germination process. In essence it is important to define more precisely how the presence of fungi leads to rupture at the crack initiation zone.

Thank you, this is a very helpful criticism to further improve the manuscript. To specifically address these points (also addressed by reviewer #4) we added new results to Figure 4 and to the main text (pages 6-8) of the revised manuscript:

(i) Figure 4g, clearly demonstrating that the fungal activity also degrades the outer pericarp (exocarp & mesocarp) in the distal region of the fruit valve, but it does not visibly affect the endocarp/lignin in the distal region where the pericarp rupture starts and the radicle emerges during germination. The degradation of the outer pericarp is therefore indeed a process which occurs in all regions and the crack initiation zone must therefore be defined by the differences in endocarp morphology and biomechanics. That this is the case is now also supported by ...

(ii) ... Figure 4h: These are the striking findings from an localised (distal versus proximal) abrasion experiment which we conducted with surface-sterilised fruit valves (no fungi). Localised removal by mechanical abrasion of the outer pericarp (leaving the endocarp intact) provided direct evidence that mechanical weakening of the DISTAL OUTER pericarp is sufficient to release pericarp-dormancy and trigger germination. Localised removal of the PROXIMAL OUTER pericarp does NOT release the pericarp-dormancy. This explicitly demonstrates that the identified novel mechanism is purely mechanical and is based on the pre-formed morphology of the distal region. This demonstrates that the only requirement for this mechanism is that the fungal activity mechanically weakens the distal pericarp by degradation of the outer layers. Reviewer #5 correctly proposes that this is "adaptive morphology enhancing germination,

facilitated by fungal degradation". The essence of this novel mechanism in this very specific region is that the fungal activity triggers a pre-formed mechanical structure to rupture.

Alternatively it could be further considered whether the exo- or mesocarp limit water uptake into the seed, and thus prevent embryo tissue expansion required for crack initiation. In this scenario the microscopy in Figure 6 becomes more incident to the story, given that the presence of these lignified zones is well documented in the Brassicaceae. The data in figure 2 appear to suggest that water uptake is not important but I was slightly unclear because of point 2 below. After fungal activity, how is pericarp permeability affected?

The fruit valve has a NPO (Natural Pericarp Opening; see Fig. 1) and therefore water uptake into the seed requires no changes in the pericarp permeability. The fungal activity is therefore not acting by affecting pericarp permeability, it triggers a completely mechanical mechanism (see above).

We apologise that Figure 2b was not very clear and also that it lacked presenting water uptake during the late stages ("water uptake associated with post-lag events" also addressed by reviewer #4). Figure 2b is now completely revised to aid clarity and to include all the phases of water uptake into the seed. The relevant information is added to the main text (results p. 6/7) and discussed (p. 11). The water uptake experiments reveal:

(i) Seeds extracted from dry fruit valves germinate with the very typical triphasic pattern of water uptake: Phase-I is imbibition, very fast and mainly driven matrix forces. Phase-II is a typical plateau phase in which the metabolism is activated and the seed prepares for the completion of germination. Phase-III is the late increase in water uptake associated with embryo elongation and the completion of germination. Based on this triphasic pattern of water uptake of the extracted (isolated) seed, it is very clear how the pericarp confers the dormancy: It mechanically prevents full phase-II level water uptake and the transition to phase-III.

(ii) If the seed is encased in the fruit valve the initial water uptake by imbibition (phase-I) is possible. The seed water content then remains at a roughly unchanged plateau level for a very long time (Fig. 2b) and is only increasing upon pericarp rupture. This plateau level seed water content is slightly, but significantly lower compared to the phase-II levels of extracted seeds. The mechanism by which the pericarp imposes the dormancy is therefore a purely mechanical restraint to further water uptake into the seed. Release of this restraint by weakening of the mechanical resistance of the distal pericarp allows further (phase-III) water uptake by the seed leading to pericarp rupture, embryo expansion growth, and the completion of fruit valve germination.

Minor points:

2. Figure 2b is hard to understand: I wonder if it can be separated to ease comprehension of the data presented. The relevance of the water uptake to the causation of germination could be discussed more clearly.

See above.

Also, in Figure 3b the symbol labels are only partly clear.

Is corrected.

3. In the discussion the potential roles of individual fungal species are discussed, but in the data presented no attempt is made to distinguish between the activities of the different fungal species. Is this discussion a little speculative without further data?

We clearly demonstrate that the fungal activity required degrades the outer pericarp cell walls and that no lignin-degrading activity is required (endocarp not affected). To further investigate the biochemical details of the specific fungal species activities is beyond the scope of this

publication. While it is an interesting follow-up research project into the details of plant cell wall degradation, it does not add anything of general interest to the discovery of the novel pericarp-imposed mechanical dormancy mechanism.

4. The proliferation of acronyms in the manuscript may hinder comprehension by a more general audience.

We have removed some acronyms not needed for the figures, e.g. PY.

5. The last two paragraphs (page 10) of the results section strays into discussion. The discussion raises obvious question is whether the fungi can be added back individually to show which are important in the fruit degradation?

We have shortened this discussion by removing details (page 10). Re-inoculation of fungi indeed demonstrates the importance of the fungal activity (see Figure 3). We however did not try adding back individually for the same reasons described above (point 3). This is in our opinion not required for demonstrating the release mechanism of the pericarp-imposed mechanical dormancy which is the novel finding of our work.

6. The manuscript title could be modified to be more informative in respect to the conclusions.

We think that our title is informative, but are of course happy to consider better proposals.

Reviewer #2 (Remarks to the Author):

The manuscript by Sperber et al. gives a detailed description on the mechanisms of seed germination in *Lepidium didymium* which species is characterized by seeds covered with impermeable valve (fruit) tissue.

The results show convincingly the involvement of fungi to reduced the permeability of the surrounding valve tissue and provides detailed information on the organization of the latter tissue linked to biomechanical analysis.

The statements of this reviewer is wrong. We do not present a study on “fungi to reduce the permeability” of an “impermeable valve (fruit)”. The fruit valve of *Lepidium didymum* has a NPO (Natural Pericarp Opening, see Fig. 1) and therefore water uptake into the seed requires no changes in the pericarp permeability. The fungal activity is therefore not acting by affecting pericarp permeability, it triggers a completely novel mechanical mechanism.

Although the data are convincing, one wonders what is the novelty of this paper because several aspects of this mechanism have been described before (see cited literature). It would be useful that also in the introduction it is clearly stated what questions were left for the scientific issue addressed here. Now these aspects sometimes only came back in the discussion.

Because this reviewer wrongly assumes that we present work on pericarp permeability (the cited literature) he/she thinks it is not novel. What we present is a novel purely mechanical mechanism of dormancy and its release by fungal activity.

The paper in general is well writing although this reviewer was a bit surprised by the sentence on page 3 about greenhouse warming. It seems that sometimes people like to have some of this modern items in all papers about ecology.

We have revised this sentence and removed the “greenhouse warming”.

The legend of fig 4 is a bit confusing. It tells that a-d are SEM pictures but this is only the case for Fig 4 b-d.

Thank you, we have improved and corrected the legend of Figure 4.

Reviewer #3 (Remarks to the Author):

This is a thorough investigation of the physical control of germination by a hard multilayered pericarp in *Lepidium didymum*, a widely distributed species of *Lepidium*. I would prefer terminology like widely distributed over cosmopolitan which means: familiar with and at ease in many different countries and cultures. I have no major complaints about the research or its interpretations. I think the paper could be condensed slightly without a loss of meaning.

Thank you for these comments. We have replaced “cosmopolitan” by widely distributed (page 4). We have also shortened the text by removing details on the various fungi (page 10) and by eliminating repetition in the discussion (page 12).

I would prefer to see paragraphs indented when the block style is not used. This makes it much easier for the reader to determine where new paragraphs begin.

We assume that this will be part of the copy-editing process when the print version is produced.

Reviewer #4 (Remarks to the Author):

I found this manuscript to be well-written and informative. It provides novel information that I feel will be of interest to persons in the field as well as those generally interested in the soil/seed microbiome. The manuscript contains a significant amount of work and I found the experiments to be properly designed and analyzed. The conclusions are generally supported by the data and the manuscript is appropriately supported with referenced material. I have a few suggestions for the authors.

Thank you for high-lighting the novelty and general importance of our interdisciplinary work. Both, the seed-fungi interaction and the underpinning biomechanics are novel mechanisms. The soil/seed microbiome is indeed a timely topic. The work presented by us is beyond this unique in that we used an interdisciplinary methodology with also including biomechanical engineering. This approach led to the discovery of a novel dormancy mechanism which is purely mechanical. In the revised version we have included results from additional experiments which further support our conclusions.

Summary – the summary does not contain any specific supporting data for conclusions stated. The summary could benefit from inclusion of some specific research data, like the percentage decrease in puncture force related to fungal association, etc.

We made small changes to the abstract, but thee is in our opinion no space for inclusion of some specific research data, like the percentage decrease in puncture force related to fungal association, etc. .

Introduction- I would prefer to see a set of well-defined objectives at the end of the introduction rather than a pre-conclusion statement.

We personally like the pre-conclusion statement.

Materials and Methods – The puncture force description indicates that a mean separation (Tukey’s test) was conducted, but only SE is reported on Fig. 5a. Even though the p-value is reported in the figure legend, I would suggest replacing the error bar with letter-represented mean separation.

We stated the p-values, but if in addition the inclusion of letters is requested we can of course add them.

I think scarification should be described in a single sentence in the manuscript rather than in supplemental information. It is important to understanding data presented in several figures and there are so many ways researchers approach scarification that I think it should be described. Also, although the treatment presented here falls under the broad heading of scarification, I think pericarp cracking would be a more descriptive term to indicate not only the treatments potential impact of water movement into the fruit, solute egress from the fruit, but also the dramatic reduction in mechanical resistance.

We removed the scarification method from the supplementary information and describe it now in the text (results page 6, and methods page 16). We also included “pericarp cracking” because it indeed is a very descriptive term.

Results and discussion –

I do not feel that the authors have fully supported their conclusion that the pericarp acts to restrict imbibition. I think that it is equally possible that the pericarp is restricting water uptake associated with post-lag events. I think this is not just semantics but is fundamental to the conclusion being drawn. Unless, the authors have reference points to the end of imbibition and initiation of lag period, it is not possible to differentiate between the pericarp restricting imbibition vs. water uptake prior to germination. Given the data provided, I think using the term water uptake is the more conservative approach because it is more inclusive.

Thank you very much for this very helpful and important comment. It triggered not only to revise the wording (“imbibition” replaced by “water uptake” throughout the text), but also to conduct an additional more conclusive water uptake experiment. These results are presented in the revised Figure 2b and show that the pericarp is indeed “restricting water uptake associated with post-lag events.” beyond imbibition. As seed physiologists we are aware of the triphasic nature of water uptake (is indeed not semantics) and have included this now (page 6).

The findings from the new water uptake experiment (Figure 2b) are described in the response to the major comment of reviewer #1 who made a similar point. We very much appreciate this criticism which helped to further reveal how the pericarp confers the mechanical dormancy.

The authors also indicate that the pericarp through solely biomechanical restraint is responsible for cessation of water uptake. However, given the fact that seeds eventually increase in fresh weight and germinate (under aseptic conditions), there could be a slow gradual change in embryo growth potential (possibly by after-ripening) that impacts gradual water uptake? Embryo growth potential measurements over the 2 to 4 months required for aseptic germination would indicate whether the embryo was involved in this phenomenon.

We show in the new Supplementary Figure 4 that the fungal activity does NOT affect the level of the phase-II water content of seeds inside fruit valves, and does NOT affect radicle size and embryo growth.

I agree with the authors that fungal activity is associated with a reduction in mechanical restraint in the pericarp. However, I am not sure they have provided definitive evidence that the portion of the pericarp responsible for imposing dormancy is due only to the parenchyma-like, thin walled meso and exocarp cells and not the endocarp cells associated with the distal portion of the fruit. I agree with the authors, that the outer pericarp is the most likely target for fungi-associated degradation as shown in Figure 4e, but I would have liked to see a non-fungal outer pericarp removal experiment to provide evidence that removal of these two tissues alone was sufficient for dormancy release. If it was not possible to physically separate these tissues, then I think it should be stated in the manuscript.

We very much liked this proposal and therefore conducted a “non-fungal outer pericarp removal experiment” for which the results are presented in Figure 4h: These are the striking findings

from an abrasion experiment which we conducted with surface-sterilised fruit valves (no fungi). Localised removal by mechanical abrasion of the outer pericarp (leaving the endocarp intact) provided direct evidence that mechanical weakening of the DISTAL OUTER pericarp is sufficient to release pericarp-dormancy and trigger germination. Localised removal of the PROXIMAL OUTER pericarp does NOT release the pericarp-dormancy. This explicitly demonstrates that the identified novel mechanism is purely mechanical and is based on the pre-formed morphology of the distal region. This demonstrates that the only requirement for this mechanism is that the fungal activity mechanically weakens the distal pericarp by degradation of the outer layers. Reviewer #5 correctly proposes that this is “adaptive morphology enhancing germination, facilitated by fungal degradation”. The essence of this novel mechanism in this very specific region is that the fungal activity triggers a pre-formed mechanical structure to rupture.

Also, pericarp sections for fruits at 0 and 35 days with and without fungal association to show that only the exo and mesocarp cells were impacted and not the distal section of the endocarp would provide further support for their conclusion. Figure 4e does show a general lack of cellular change in endocarp cells, but I was particularly interested in any localized changes in the cells over the radicle indicated as iPBZ in Figure 6. It is quite plausible that these initial break cells are also involved in mechanical restraint and could be locally impacted by fungal enzymes. Did the authors take sections in the distal region similar to those in Fig. 4e that could provide additional information?

In Figure 4g we now present the effects of the fungal activity on the DISTAL pericarp. Figure 4g clearly demonstrates that the fungal activity also degrades the outer pericarp (exocarp & mesocarp) in the distal region of the fruit valve, but it does not visibly affect the endocarp/lignin in the distal region where the pericarp rupture starts and the radicle emerges during germination. The degradation of the outer pericarp is therefore indeed a process which in the DISTAL pericarp weakens the structure. See above for the “non-fungal outer pericarp removal experiment” for which the results are presented in Figure 4h. The distal endocarp was not affected by these treatments (fungi or abrasion) and this further supports our conclusion about the specific pre-formed endocarp morphology and biomechanics which is the underpinning mechanical dormancy mechanism.

These previous comments are not meant to be critical of the conclusion that fungal activity is associated with reduction of pericarp mechanical restraint, only the specific localization of that impact on the out portion of the pericarp alone.

Yes indeed, we now explicitly demonstrate the impact of the distal outer pericarp.

The discussion is mostly a restatement of results and I think it could be easily condensed.

We condensed the discussion by removal of the restatement of results on page 11, 12 and 14.

Reviewer #5 (Remarks to the Author):

This paper presents the results of detailed studies on the biomechanical and ecophysiological characteristics of seed morphology and germination behavior of *Lepidium*, which exhibits mechanical dormancy where the hard pericarp prevents seeds from fully imbibing water from the surrounding environment. A novel finding is that fungi appear to play a key role in degrading targeted pericarp tissues which allows full imbibition and germination to occur. A characteristic group of seed-inhabiting fungi were identified and shown to be capable of degrading pericarp tissues. This conclusion was aided by comparisons of germination of seed with the pericarp removed (eliminating biomechanical constraints on imbibition and germination) and by surface sterilizing seeds to prevent fungal colonization of pericarp structures. In addition, detailed biomechanical and microscopic studies showed that specific regions of the pericarp differ in such a manner that a crack or rupture point originates in a very specific region, consistent with the idea of adaptive morphology enhancing seed germination, and facilitated by fungal degradation.

The discovered mechanical dormancy is indeed a prime example for “adaptive morphology enhancing seed germination, and facilitated by fungal degradation”. Thank you for this excellent summary of our work. The mechanism how hard coverings determine the ecophysiological behaviour of diaspores were a riddle until now.

Overall, I found this to be an interesting and well written paper with excellent imagery. Fig. 3A especially shows the interaction of fungi with pericarp-associated dormancy. However, I question the level of general interest in this paper in a broad-based journal like Nature Communications compared to a more specialized botanical journal where the detailed morphological and biomechanical studies might be more highly appreciated. While the role of fungi in the germination process is novel, it is unclear whether most other soil fungi could accomplish the same pericarp degradation or whether the five identified taxa are consistently associated with *Lepidium* across its worldwide distribution. Thus, while the paper and study are very well done, it is not clear to me whether the paper would be of interest beyond a small group of specialists.

We do not agree on this statement, but are convinced that our unique interdisciplinary approach which led to the discovery of a novel dormancy mechanism is of broad interest and will trigger discussion across disciplines. Our view is supported by the other reviewers, e.g. “will likely be wide general interest in the new aspect of plant-fungi symbiotic behaviour” (Reviewer #1) and “provides novel information that I feel will be of interest to persons in the field as well as those generally interested in the soil/seed microbiome.” (Reviewer #4).

Discussion – *Lepidium* has a worldwide distribution so it occurs in many climatic conditions and great variation in precipitation, temperature, etc. where fungal activity would also differ. So it was unclear to me how this one mechanism of breaking dormancy could operate successfully over such a wide range of conditions.

The reason for this is that this novel dormancy is (a) based on a purely mechanical mechanism to impose the dormancy, namely by restricting water uptake into the seed which operates at any temperature and climatic condition. And (b) that its release is mediated by common ascomycetes which are globally distributed and present in all climatic regions where *Lepidium didymium* is present (Supplementary Figure 7). It is therefore a novel mechanism to spread germination in time which solely depends on the presence and abundance of the common fungi to degrade the outer pericarp. These common fungi are adapted to the various climates to proliferate depending on the (seasonal) humidity/temperature conditions. It is therefore a general mechanism and not a specific microbe-host interaction depending on strains specifically adapted to the plant host. We think that variation in precipitation, temperature, etc. where fungal activity would differ between climates is what provides the climatic adaptation of this mechanism.

Table 1 – The table lists common fungi but what about other fungi? What fraction of OTUs did these five taxa represent? The text says 5-6 common fungi but little detail is provided. Would any type of generic soil fungi have the same activity on pericarp degradation vs. a group of highly coevolved, seed-associated fungi?

The method used identifies all fungi (Ascomycota, Basidiomycota, early dividing lineages) as evident from the Table 1 caption and method citation. We identified these 5-6 species of common ascomycetes, but did not identify any basidiomycetes or fungi from early dividing lineages. *Cladosporium sp.* was most abundant, and the identified fungi represent the major fruit-associated fungi. Other soil fungi may be able to degrade the pericarp, but we did not perform any experiments with soil fungi. To our knowledge there is no evidence for any highly coevolved, seed-associated fungi affecting dormancy. The identified fungi and the nature of the discovered purely mechanical dormancy mechanism does not require any highly coevolved, seed-associated fungi.

REVIEWERS' COMMENTS:

Reviewer #1 (Remarks to the Author):

The authors deal effectively with my previous concerns. My only comment is that in the new version Figure 5 seems somewhat incidental, as all the relevant morphological aspects of the seeds are shown clearly in figure 3.

REVIEWERS' COMMENTS:

Reviewer #1 (Remarks to the Author): The authors deal effectively with my previous concerns. My only comment is that in the new version Figure 5 seems somewhat incidental, as all the relevant morphological aspects of the seeds are shown clearly in figure 3.

We assume that this reviewer refers to the Figure 6 in the revised version. It is our opinion that this Figure is not identical and shows different morphological aspects of the fruit coat (not seeds) and therefore new results not shown in any other figure. We therefore would prefer if the figure remains part of the article, but if the Editor insists we can also agree on moving this figure to the Supplementary File. We leave this decision to the discretion of the Editor.